# ACAT: ADVERSARIAL COUNTERFACTUAL ATTENTION FOR CLASSIFICATION AND DETECTION IN MEDICAL IMAGING

## ABSTRACT

In some medical imaging tasks and other settings where only small parts of the image are informative for the classification task, traditional CNNs can sometimes struggle to generalise. Manually annotated Regions of Interest (ROI) are sometimes used to isolate the most informative parts of the image. However, these are expensive to collect and may vary significantly across annotators. To overcome these issues, we propose a method to generate ROIs via saliency maps, obtained from adversarially generated counterfactual images. With this method, we are able to isolate the area of interest in brain and lung CT scans without using any manual annotations. Our saliency maps, in the task of localising the lesion location out of 6 possible regions, obtain a score of $65.05\%$ on brain CT scans, improving the score of $61.29\%$ obtained with the best competing method. We then employ the saliency maps in a framework that refines a classifier pipeline; in particular, the saliency maps are used to obtain soft spatial attention masks that modulate the image features at different scales. We refer to our method as *Adversarial Counterfactual Attention* (ACAT). ACAT increases the baseline classification accuracy of lesions in brain CT scans from $71.39\%$ to $72.55\%$ and of COVID-19 related findings in lung CT scans from $67.71\%$ to $70.84\%$ and exceeds the performance of competing methods.

## 1 INTRODUCTION

In computer vision classification problems, it is often assumed that an object that represents a class occupies a large part of an image. However, in other image domains, such as medical imaging or histopathology, only a small fraction of the image contains information that is relevant for the classification task (Kimeswenger et al., 2019). With object-centric images, using wider contextual information (e.g. planes fly in the sky) and global features can aid the classification decision. In medical images, variations in parts of the image away from the local pathology are often normal, and using any apparent signal from such regions is usually spurious and unhelpful in building robust classifiers. Convolutional Neural Networks (CNNs) (Krizhevsky et al., 2012; He et al., 2016; Szegedy et al., 2017; Huang et al., 2017a) can struggle to generalise well in such settings, especially when training cannot be performed on a very large amount of data (Pawlowski et al., 2019). This is at least partly because the convolutional structure necessitates some additional 'noisy' statistical response to filters away from the informative 'signal' regions. Because the 'signal' response region is small, and the noise region is potentially large, this can result in low signal to noise in convolutional networks, impacting performance.

To help localisation of the most informative parts of the image in medical imaging applications, *Region Of Interest* (ROI) annotations are often collected (Cheng et al., 2011; Papanastasopoulos et al., 2020). However, these annotations require expert knowledge, are expensive to collect, and opinions on ROI of a particular case may vary significantly across annotators (Grünberg et al., 2017). Alternatively, attention systems could be applied to locate the critical regions and aid classification. Previous work has explored the application of attention mechanisms over image features, either aiming to capture the spatial relationship between features (Bell et al., 2016; Newell et al., 2016; Santoro et al., 2017), the channel relationship (Hu et al., 2018) or both (Woo et al., 2018; Wang et al., 2017). Other authors employed self-attention to model non-local properties of images (Wang

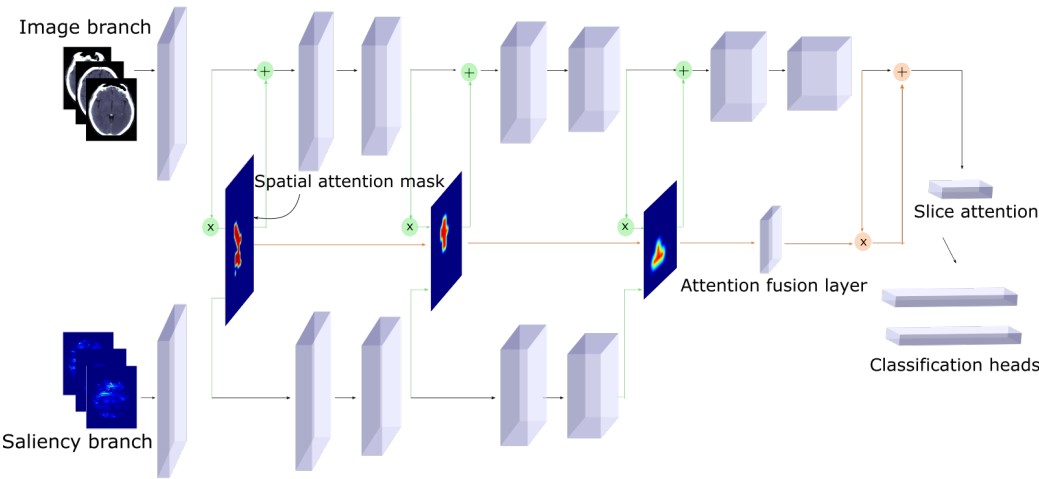

Figure 1: Architecture of the framework proposed for 3D volumes. The slices of each volume are first processed separately and then combined by applying an attention module over the slices. For each volume we also consider as input the corresponding saliency map. From the saliency branch, we obtain soft spatial attention masks that are used to modulate the image features. The salient attention modules capture information at different scales of the network and are combined through an attention fusion layer to better inform the final classification.

et al., 2018; Zhang et al., 2019). However, in our experiments, attention methods applied on the image features failed to improve the baseline accuracy in brain and lung CT scans classification. Other authors employed saliency maps to promote the isolation of the most informative regions during training of a classification network. They sometimes employed target ground-truth maps to generate these saliency maps (Murabito et al., 2018). Moreover, by fusing salient information with the image branch at a single point of the network (Murabito et al., 2018; Flores et al., 2019; Figueroa-Flores et al., 2020), these approaches may miss important data. Indeed, when the signal is low, key information could be captured by local features at a particular stage of the network, but not by features at a different scale.

We propose to use counterfactual images, acquired with a technique similar to adversarial attacks (Huang et al., 2017b), as a means to acquire saliency maps which highlight useful information about a particular patient's case. In general, counterfactual examples display the change that has to be applied to the input image for the decision of a black-box model to change. Our method achieves good isolation of the area of interest, without requiring any annotation masks. In particular, for generating counterfactual examples, we employ an autoencoder and a trained classifier to find the minimal movement in latent space that shifts the input image towards the target class, according to the output of the classifier. These saliency maps can also be used in a classification pipeline, as shown in Figure 1, to obtain soft spatial attention masks that modulate the image features. To capture information at different scales, the attention masks are computed from the saliency features at different stages of the network and also combined through an attention fusion layer in order to better inform the final decision of the network.

The main contributions of this paper are the following: 1) we introduce a method to generate counterfactual examples, from which we obtain saliency maps that outperform competing methods in isolating small areas of interest in large images, achieving a score of $65.05\%$ in the task of localising the lesion location out of 6 possible regions on brain CT scans (vs. $61.29\%$ obtained with the best competing method), 2) we propose ACAT, a framework that employs these saliency maps as attention mechanisms at different scales and show that it improves the baseline classification accuracy in two medical imaging tasks (from $71.39\%$ to $72.55\%$ on brain CT scans and from $67.71\%$ to $70.84\%$ in lung CT scans), 3) we show how ACAT can also be used to evaluate saliency generation methods.

## 2  RELATED WORK

While a complete overview of the methods used to generate saliency maps, counterfactual examples and adversarial attacks is out of the scope of this work, we briefly summarise some of the approaches most commonly used in medical imaging.

**Saliency maps** Saliency maps are a tool often employed by researchers for post-hoc interpretability of neural networks. They help to interpret CNN predictions by highlighting pixels that are important for model predictions. Simonyan et al. (2013) compute the gradient of the score of the class of interest with respect to the input image. The Guided Backpropagation method (Springenberg et al., 2014) only backpropagates positive gradients, while the Integrated Gradient method (Sundararajan et al., 2017) integrates gradients between the input image and a baseline black image. In SmoothGrad (Smilkov et al., 2017), the authors propose to smooth the gradients through a Gaussian kernel. Grad-CAM (Selvaraju et al., 2017) builds on the Class Activation Mapping (CAM) (Zhou et al., 2016) approach and uses the gradients of the score of a certain class with respect to the feature activations of the last convolutional layer to calculate the importance of the spatial locations.

**Counterfactuals for visual explanation** Methods that generate saliency maps using the gradients of the predictions of a neural network have some limitations. Some of these methods have been shown to be independent of the model parameters and the training data (Adebayo et al., 2018; Arun et al., 2021) and not reliable in detecting the key regions in medical imaging (Eitel et al., 2019; Arun et al., 2021). For this reason, alternative methods based on the generation of counterfactuals for visual explanation have been developed. They are usually based on a mapping that is learned between images of multiple classes to highlight the areas more relevant for the class of each image. The map is modeled as a CNN and is trained using a Wasserstein GAN (Baumgartner et al., 2018) or a Conditional GAN (Singla et al., 2021). Most close to our proposed approach to generate counterfactuals, is the latent shift method by Cohen et al. (2021). An autoencoder and classifier are trained separately to reconstruct and classify images respectively. Then, the input images are perturbed to create $\lambda$-shifted versions of the original image that increase or decrease the probability of a class of interest according to the output of the classifier.

**Saliency maps to improve classification and object detection** Previous work has tried to incorporate saliency maps to improve classification or object detection performance in neural networks. Ren et al. (2013) used saliency maps to weigh features. Murabito et al. (2018) introduced SalClassNet, a framework consisting of two CNNs jointly trained to compute saliency maps from input images and using the learned saliency maps together with the RGB images for classification tasks. In particular, the saliency map generated by the first CNN is concatenated with the input image across the channel dimension and fed to the second network that is trained on a classification task. Flores et al. (2019) proposed to use a network with two branches: one to process the input image and the other to process the corresponding saliency map, which is pre-computed and given as input. The two branches are fused through a modulation layer which performs an element-wise product between saliency and image features. They observe that the gradients which are back-propagated are concentrated on the regions which have high attention. In (Figueroa-Flores et al., 2020) the authors use the same modulation layer, but replace the saliency branch that was trained with pre-computed saliency images with a branch that is used to learn the saliency maps, given the RGB image as input.

**Adversarial examples and adversarial training** Machine learning models have been shown to be vulnerable to adversarial examples (Papernot et al., 2016). These are created by adding perturbations to the inputs to fool a learned classifier. They resemble the original data but are misclassified by the classifier (Szegedy et al., 2013; Goodfellow et al., 2014). Approaches proposed for the generation of adversarial examples include gradient methods (Kurakin et al., 2018; Moosavi-Dezfooli et al., 2016) and generative methods (Zhao et al., 2017). In Qi et al. (2021), the authors propose an adversarial attack method to produce adversarial perturbations on medical images employing a loss deviation term and a loss stabilization term. In general, adversarial examples and counterfactual explanations can be created with similar methods. Adversarial training, in which each minibatch of training data is augmented with adversarial examples, promotes adversarial robustness in classifiers (Madry et al., 2017). Tsipras et al. (2018) observe that gradients for adversarially trained networks are well aligned with perceptually relevant features. However, adversarial training usually also decreases the accuracy of the classifier (Raghunathan et al., 2019; Etmann et al., 2019).

## 3 METHODS

We wish to automatically generate and make use of RoI information in the absence of hand-labelled annotations. The approach we take can be considered similar to adversarial training, in the sense that we obtain saliency maps from adversarially generated counterfactual images and use them to train a classifier. However, given $f$ classification network and $x^i$ input image, directly modifying the image using $\frac{\partial f(x^i)}{\partial x^i}$, as in common adversarial attack approaches, would distort the image through imperceptible modifications that fool the network. Instead, the use of an autoencoder keeps the image on the data manifold and leads to pixel changes that are semantically meaningful. Moreover, we are able to employ these adversarially generated saliency maps in a framework that improves, rather than reduces, the classification accuracy.

### 3.1 GENERATION OF COUNTERFACTUAL EXAMPLES

In order to detect regions of interest in medical images, we generate counterfactual examples for each datum and use the difference with the original image to generate a saliency map highlighting important information. In particular, given a dataset $\mathcal{D} = (x^i; i = 1, 2, \ldots, N_D)$ of size $N_D$ consisting of input images $x^i$, along with corresponding class labels $\mathcal{T} = (y^i; i = 1, 2, \ldots, N_D)$, counterfactual explanations describe the change that has to be applied to an input for the decision of a black-box model to flip. Let $f$ be a neural network that outputs a probability distribution over classes, and let $\hat{y}^i$ be the class designated maximum probability by $f$. A counterfactual explanation displays how $x^i$ should be modified in order to be classified by the network as belonging to a different class of interest $\bar{y}^i$ (counterfactual class). In order to generate saliency maps, we can consider the difference between the original image and the counterfactual image of the opposite class. For example, to compute the saliency map of a brain scan with a stroke lesion, we could generate a counterfactual example that is classified by $f$ as not having a stroke lesion. In this way, we are able to visualise the pixels with the biggest variation between the two samples, which are the most important for the classification outcome. However, when using saliency maps to improve the classification capability of our network, at test time we don't have access to class labels. For this reason, to compute saliency maps in a class-agnostic way, we consider the counterfactual examples of both classes (positive and negative) and then compute the absolute difference between the original image and each counterfactual image to get two attribution maps. These are then normalised in $[0, 1]$ and averaged to obtain the final saliency map that can be used in the classification pipeline.

As discussed, gradient-based counterfactual changes to image pixels can just produce adversarial attacks. We alleviate this by targeting gradients of a latent autoencoder. Therefore, in addition to the network $f$, trained to classify images in $\mathcal{D}$, we exploit an autoencoder, trained to reconstruct the same inputs. $x^j \in \mathcal{D}$ can be mapped to latent space through the encoder $E$: $E(x^j) = z^j$. This can then be mapped back to image space via decoder D: $x'^j = D(z^j)$. Suppose without loss of generality that the counterfactual example we are interested in belongs to a single target class. The neural network can be applied to this decoder space, we denote the output of $f(D(z^j))$ as a normalised probability vector $d(z^j) = (d_1(z^j), \ldots, d_k(z^j)) \in \mathbb{R}^K$, where K is the number of classes. Suppose that $f(x^j)$ outputs maximum probability for class $l$ and we want to shift the prediction of $f$ towards a desired class $m$, with $l, m \in \mathbb{N} : l, m \in [1, K]$. To do so, we can take gradient steps in the latent space of the autoencoder from initial position $z^j$ to shift the class distribution towards the desired target vector $t = (t_1, \ldots, t_k) \in \mathbb{R}^K$, where $t_i = \mathbf{1}_{i=m}$, for $i = 1, \ldots, K$ . In order to do so, we would like to minimise the cross-entropy loss between the output of our model, given $D(z^j)$ as input, and the target vector. I.e. we target

$$L(d(z^j), t) = -\sum_{k=1}^{K} t_k \log(d_k(z^j)). \tag{1}$$

Moreover, we aim to keep the counterfactual image as close as possible to the original image in latent space, so that the transformation only captures changes that are relevant for the class shift. Otherwise, simply optimising Eq. (1) could lead to substantial changes in the image that compromise its individual characteristics. Therefore, we also include, as part of the objective, the $L_1$ norm between the latent spaces of the original image $x^j$ and the counterfactual image: $||z - E(x^j)||_{L_1}$. Putting things together, we wish to find the minimum of the function:

$$g(z) = L(d(z), t) + \alpha ||z - E(x^j)||_{L_1} \tag{2}$$

where $\alpha$ is a hyperparameter that was set to $100$ in our experiments. We can minimise this function by running gradient descent for a fixed number of steps (20 in our experiments). Then, for the minimizer of Eq. (2), denoted by $z'$, the counterfactual example is given by $D(z')$.

By defining an optimisation procedure over the latent space that progressively optimises the target classification probability of the reconstructed image, we are able to explain the predictions of the classifier and obtain adequate counterfactuals. A bound on the distance between original and counterfactual images in latent space is also important to keep the generated samples within the data manifold.

### 3.1.1 DIFFERENCE FROM THE LATENT SHIFT METHOD

Following the same notation as before, given an input image $x^k$, with latent space $z^k = E(x^k)$, Cohen et al. (2021) propose a method to generate counterfactuals by creating perturbations of the latent space in the following way: $z^k_\lambda = z^k + \lambda \frac{\partial f(D(z^k))}{\partial z^k}$, where $\lambda$ is a sample-specific hyperparameter that needs to be found by grid search. These representations can be used to create $\lambda$-shifted versions of the original image: $x^k_\lambda = D\left(z^k_\lambda\right)$. For positive values of $\lambda$, the new image $x^k_\lambda$ will produce a higher prediction, while for negative values of $\lambda$, it will produce a lower prediction. Depending on the landscape of the loss, the latent shift approach may be unsuitable to reach areas close to a local minimum and fail to correctly generate counterfactuals. The reason is that this method can be interpreted as a one-step gradient-based approach, trying to minimise the loss of $f(D(z^k))$ with respect to the target probability for the class of interest, with one single step of size $\lambda$ in latent space. To solve this issue, we propose an optimisation procedure employing small progressive shifts in latent space, rather than a single step of size $\lambda$ from the input image. In this way, the probability of the class of interest converges smoothly to the target value. We show examples of the failure modes of the latent shift method, where the probability of the class of interest does not converge to the target value, that are fixed by our progressive optimisation in Appendix E. Another issue of the latent shift method is that it doesn't introduce a bound on the distance between original and counterfactual images. Therefore, the generated samples are not always kept on the data manifold and may differ considerably from the original image. To solve this issue, we add a regularisation term that, limiting the move in latent space, ensures that the changes that we observe can be attributed to the class shift and the image doesn't lose important characteristics.

### 3.2 SALIENCY BASED ATTENTION

Once we obtain a counterfactual example with our method, we then use it to obtain a saliency map, which we can inject into a classification network to learn better local features and improve the classification accuracy. We do so through a saliency branch, which has attention modules that learn how to handle the salient information coming into the system and use it to obtain soft spatial attention masks that modulate the image features. In particular, with reference to Figure 1, we consider a network with two branches, one for the original input images and the other for the corresponding saliency maps, which are pre-computed and fixed during training of the network. Given $S^i \in \mathbb{R}^{C \times H \times W}$ features of the saliency branch at layer $i$, we first pool the features over the channel dimension to obtain $S^i_p \in \mathbb{R}^{1 \times H \times W}$. Both average or max-pooling can be applied. However, in preliminary experiments we found max-pooling to obtain a slightly better performance. A convolution with $3 \times 3$ filters is applied on $S^i_p$, followed by a sigmoid activation, to obtain soft spatial attention masks based on salient features $S^i_s \in \mathbb{R}^{1 \times H \times W}$. Finally, the features of the image branch at layer $i$: $F^i \in \mathbb{R}^{C \times H \times W}$ are softly modulated by $S^i_s$ in the following way:

$$F^i_o = F^i \odot S^i_s \tag{3}$$

where $\odot$ is the Hadamard product, in which the spatial attention values are broadcasted along the channel dimension, and $F^i_o$ are the modulated features for the $i-th$ layer of the image branch. We also introduce skip connections between $F^i$ and $F^i_o$ to prevent gradient degradation and distill information from the attention features, while also giving the network the ability to bypass spurious signal coming from the attention mask. Therefore, the output of the image branch at layer $i$, is given by: $G^i = F^i + F^i_o$

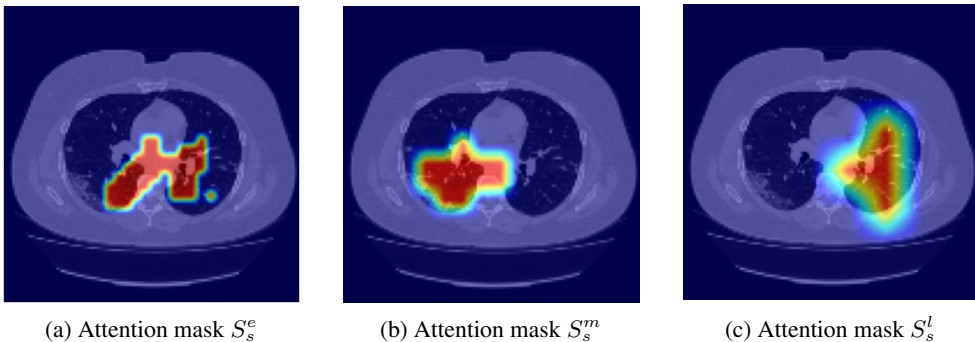

(a) Attention mask $S_s^e$        (b) Attention mask $S_s^m$        (c) Attention mask $S_s^l$

Figure 2: Spatial attention masks obtained after early (a), middle (b) and late (c) convolutional layers that focus on different parts of the image. They are weighted by the attention fusion layer depending on their importance for the classification outcome.

The attention mask not only modulates the image features during a forward pass of the network, but can also cancel noisy signal coming from the image features during backpropagation. Indeed, if we compute the gradient of $G^i$ with respect to the image parameters $\theta$, we obtain:

$$\frac{\partial G^i(\theta; \eta)}{\partial \theta} = \frac{\partial [F^i(\theta) + F^i(\theta) \odot S_s^i(\eta)]}{\partial \theta} = \frac{\partial F^i(\theta)}{\partial \theta} S_s^i(\eta) \tag{4}$$

where $\eta$ are the saliency parameters.

### 3.2.1 FUSION OF ATTENTION MASKS

Previous work attempting to exploit saliency maps in classification tasks, has fused salient information with the image branch at a single point of the network, either directly concatenating attribution maps with the input images (Murabito et al., 2018) or after a few layers of pre-processing (Flores et al., 2019; Figueroa-Flores et al., 2020). On the other hand, we position our salient attention modules at different stages of the network, in order to capture information at different scales. This is particularly important in low signal-to-noise tasks, where the key information could be captured by local features at a particular stage of the network, but not by features at a different scale. For this reason, we use three attention modules, after early, middle and late layers of the network. Given $S_s^e$, $S_s^m$ and $S_s^l$ the corresponding spatial attention masks, we also reduce their height and width to $H'$ and $W'$ through average pooling, obtaining $S_{s,p}^e$, $S_{s,p}^m$ and $S_{s,p}^l$ respectively. Then, we concatenate them along the channel dimension, obtaining $S_{s,p} \in \mathbb{R}^{3 \times H' \times W'}$. An attention fusion layer $L_f$ takes $S_{s,p}$ as input and generates a fused spatial mask $S_f \in \mathbb{R}^{1 \times H' \times W'}$ by weighting the three attention masks depending on their relative importance. This final attention mask is applied before the fully-connected classification layers, so that if critical information was captured in early layers of the network, it can better inform the final decision of the network. In practice, $L_f$ is implemented as a $1 \times 1$ convolution.

## 4 EXPERIMENTS

### 4.1 DATA

We performed our experiments on two datasets: IST-3 (Sandercock et al., 2011) and MosMed (Morozov et al., 2020). Both datasets were divided into training, validation and test sets with a 70-15-15 split and three runs with different random seeds were performed. More details about the data are provided in Appendix A.

### 4.2 EXPERIMENTAL SETUP

The baseline model for the classification of stroke lesions in CT scans of the brain employs the same base multi-task learning (MTL) architecture of Anonymous Author (s), while for classification

of lung CT scans, we employed a ResNet-50 architecture (with 4 convolutional blocks). Further details about the architectures are provided in Appendix B. In our framework, the attention branches follow the same architecture of the baseline architectures (removing the classification layers). In the MTL model, the attention layers are added after the first, third and fifth convolutional layer. For the ResNet architecture, attention modules are added after each one of the first three convolutional blocks. The attention fusion layer is always placed after the last convolutional layer of each architecture. Moreover, instead of averaging the slices of each scan, in our framework we consider an attention mask over slices. This is obtained from image features by considering an MLP with one hidden layer. The hidden layer is followed by a leaky ReLU activation and dropout with $p = 0.1$. After the output layer of the MLP, we apply a sigmoid function to get the attention mask. Further training details are provided in Appendix C.

### 4.3 EVALUATION OF SALIENCY MAPS

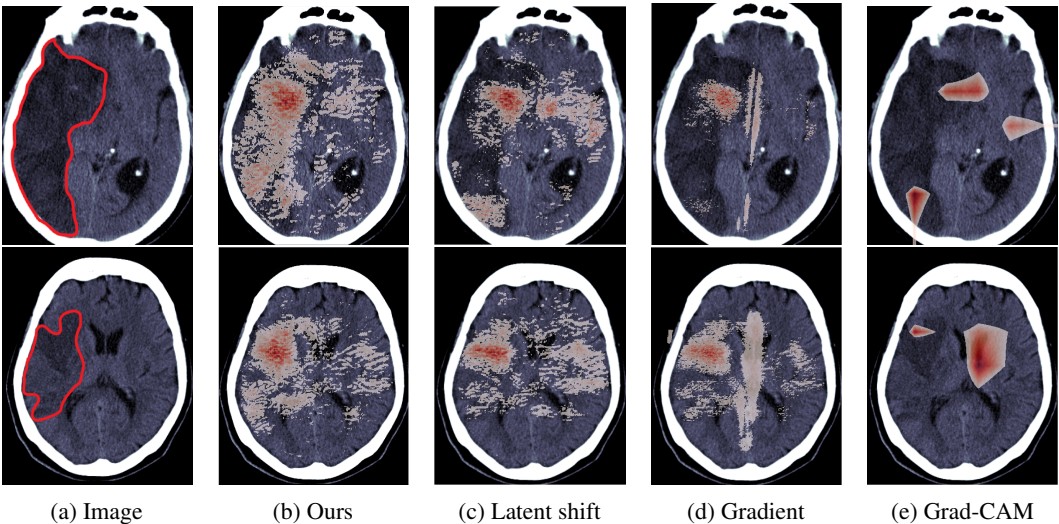

| (a) Image | (b) Ours | (c) Latent shift | (d) Gradient | (e) Grad-CAM |

Figure 3: (a) Ischaemic stroke lesion appears darker than normal brain. Sample saliency maps averaged over slices obtained with our approach (b), the latent shift method (c), the Gradient method (d) and Grad-Cam (e).

We evaluate quantitatively how the saliency maps generated with our approach described in Section 3.1, the latent shift method (Cohen et al., 2021), the gradient method (Simonyan et al., 2013) and Grad-CAM (Selvaraju et al., 2017) are able to detect the areas related to the stroke lesion. The maps were created employing the baseline model and positive scans which were not used during training. In particular, we generated negative counterfactuals with our approach and the latent shift method and computed the difference between the original image and the generated images to obtain the saliency maps. Grad-CAM is applied using the last convolutional layer of the network. The lesion location, which is used for evaluation, but is not known to the network, is one of the 6 classes: MCA left, MCA right, ACA left, ACA right, PCA left, PCA right. The attribution maps are evaluated as in Zhang et al. (2018), with the formula: $S = \frac{Hits}{Hits + Misses}$. A hit is counted if the pixel with the greatest value in each CT scan lies in the correct region, a miss is counted otherwise. The saliency maps generated with our approach obtain the highest average score of $65.05\%$ (with 2.03 standard error), improving the scores of $58.39\%$ (2.00) and $61.29\%$ (2.06) obtained with the latent shift and the gradient methods respectively. Grad-CAM has the worst score, with $11.67\%$ (1.28). Sample saliency maps are showed in Figure 3 with a red color map. The red arrows indicate the lesion regions, which appear as a 'shaded' area in the scans.

Furthermore, ACAT improves the lesion detection capabilities of saliency maps further. Indeed, if we re-compute the saliency maps with our approach and using ACAT as classifier to generate the counterfactuals, we obtain a score of $68.55\%$ (1.94), without using the class labels. In fact, the saliency maps are generated by averaging the absolute differences between the original image and the counterfactual examples of both classes (positive and negative).

## 4.4 CLASSIFICATION RESULTS

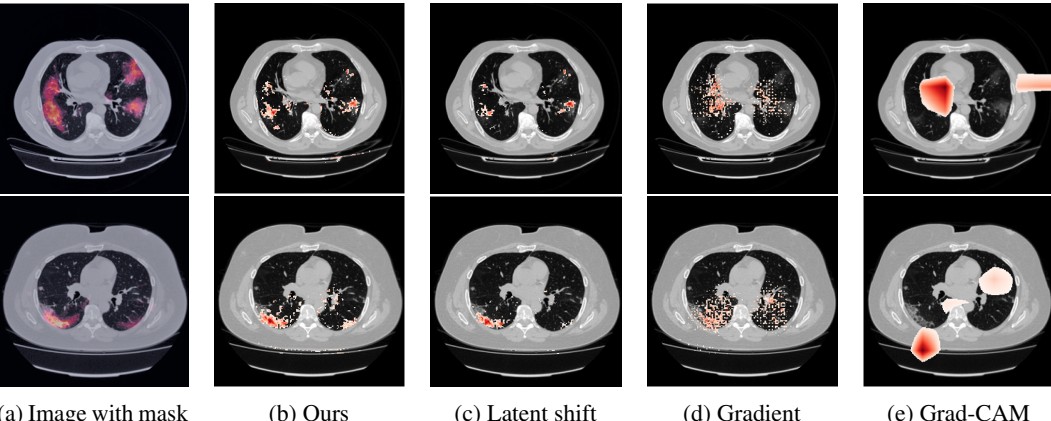

| (a) Image with mask | (b) Ours | (c) Latent shift | (d) Gradient | (e) Grad-CAM |

Figure 4: Input image with masks depicting regions of interests (a) and saliency maps averaged over slices obtained with our approach (b), the latent shift method(c), the Gradient method (d) and Grad-Cam (e)

We compare the proposed framework with competing methods incorporating saliency maps into the classification pipeline, methods employing attention from the input images and the baseline model trained without saliency maps on the classification of brain and lung CT scans. In the former case, the possible classes are: no lesion, lesion in the left half of the brain, lesion in the right half of the brain or lesion in both sides. In the latter case, we perform binary classification between scans with or without COVID-19 related findings. In methods where saliency maps are needed, for a fair comparison of the different frameworks, we always compute them with our approach. We compare our method with saliency-modulated image classification (SMIC) (Flores et al., 2019), SalClassNet (Murabito et al., 2018), hallucination of saliency maps (HSM) (Figueroa-Flores et al., 2020), spatial attention from the image features (SpAtt) and self-attention (SeAtt). Implementation details are provided in Appendix D.

As we can observe in Table 1, our approach improves the average classification accuracy of the baseline from 71.39% to 72.55% on IST-3 and from 67.71% to 70.84% on MosMed. Our framework is also the best performing in both cases. SMIC performs slightly worse than the baseline on IST-3 (with 70.85% accuracy) and better on MosMed (with 69.27% accuracy). HSM is close to the baseline results on IST-3 but worse than the baseline on MosMed, while SalClassNet is worse than the baseline on both tasks. The methods incorporating attention from the image features have also similar or worse performance than the baseline, highlighting how the use of attention from the saliency maps is key for the method to work. While it is easier to detect large stroke lesions, these can also be detected easily by humans. For this reason, we aim to test the capabilities of these models to flag scans with very subtle lesions. In order to do so, we evaluate their classification accuracy by infarct size (IS). As

Table 1: Average test accuracy (and standard error) over 3 runs on the classification of brain (IST-3) and lung (MosMed) CT scans. Our framework, ACAT, outperforms competing methods that employ saliency maps to aid classification. Methods using attention from the image features have performance similar or worse than the baseline

|  | IST-3 | MosMed |
|---|---|---|
| Baseline | 71.39% (0.23) | 67.71% (3.48) |
| SMIC | 70.85% (0.63) | 69.27% (1.13) |
| SalClassNet | 69.43% (1.81) | 62.50% (2.66) |
| HSM | 71.38% (0.94) | 65.63% (1.28) |
| SpAtt | 70.96% (0.10) | 66.67% (2.98) |
| SeAtt | 71.23% (0.10) | 67.71% (1.70) |
| ACAT (Ours) | **72.55**% (0.82) | **70.84**% (1.53) |

Table 2: Test accuracy by infarct size. Our framework, ACAT, improves the performance of competing methods in the detection of scans with no infarct lesion, small and medium lesions (size 1-2)

|  | No Lesion | IS-1 | IS-2 | IS-3 | IS-4 |
|---|---|---|---|---|---|
| Baseline | 81.41% | 23.66% | 54.16% | **72.09%** | 87.74% |
| SMIC | 79.24% | 25.55% | 54.82% | 65.71% | **88.36%** |
| SalClassNet | 76.71% | 29.24% | 54.48% | 64.95% | 82.71% |
| HSM | 80.37% | 27.28% | 53.86% | 71.60% | 89.10% |
| SpAtt | 82.56% | 21.33% | 51.58% | 67.86% | 86.77% |
| SeAtt | 83.49% | 27.03% | 52.05% | 65.54% | 84.42% |
| ACAT (Ours) | **84.30%** | **30.23%** | **55.02%** | 68.67% | 84.93% |

we can observe in Table 2 our approach obtains the best classification performance on the scans with no infarct lesion, as well as small and medium lesions (size 1-2). This confirms how our saliency based attention mechanism promotes the learning of local features that better detect subtle areas of interest.

### 4.5 ABLATION STUDIES

We compare the performance of ACAT when saliency maps obtained with different approaches are employed. When using saliency maps obtained with our approach we obtain the highest accuracy of $72.55\%$ (0.72). The relative ranking of the saliency generation approaches is the same that was obtained with the evaluation of saliency maps with the score presented in Section 4.3, with the gradient method obtaining $72.16\%$ (0.88) accuracy, the latent shift method $72.04\%$ (1.07) and Grad-CAM $69.42\%$ (1.19).

### 4.6 ACAT IS NOT RANDOM REGULARISATION

We employed dropout to test if the improvements obtained with ACAT are only due to regularization effects that can be replicated by dropping random parts of the image features. In particular, we employed dropout with different values of $p$ on the image features at the same layers where the attention masks are applied in ACAT. The accuracy obtained was lower than in the baseline models. In particular, we obtained $68.71\%, 68.36\%$ average accuracy on IST-3 for $p = 0.2, 0.6$ respectively (vs $71.39\%$ of the baseline) and $53.13\%, 58.86\%$ accuracy on MosMed for the same values of $p$ (vs $67.71\%$ of the baseline). The results suggests that spatial attention masks obtained from salient features in ACAT are informative and the results obtained with ACAT cannot be replicated by random dropping of features.

## 5 CONCLUSION

In this work, we proposed a way to generate saliency maps from adversarially generated counterfactual images that capture small areas of interest in low signal-to-noise samples. We employed these attention maps to improve classification accuracy in two medical imaging tasks (IST-3 and MosMed) by obtaining soft attention masks from salient features at different scales. These attention masks modulate the image features and can cancel noisy signal coming from the image features. They are also weighted by an attention fusion layer in order to better inform the classification outcome. Next, we showed how our framework could also be used to rank saliency maps. A possible limitation of our approach is that a baseline model is needed to compute the attribution masks that are later employed during the training of our framework. However, we believe that this approach could still fit in a normal research pipeline, as simple models are often implemented as a starting point and for comparison with newly designed approaches. While our approach has been tested only on brain and lung CT scans, we believe that it can generalise to many other medical imaging tasks and we leave further testing for future work.

## 6 ETHICS STATEMENT

Several countries are experiencing a lack of radiologists (Dall, 2018) compared to the amount of patients that need care. This can lead to several undesirable consequences, such as delays in diagnosis and subsequent treatment. Machine learning tools that automate some clinically relevant tasks and provide assistance to doctors, can lower the workload of physicians and improve the standard of care. However, many of these are black-box models and require ROI masks, which have to be annotated by specialists, to be trained. On the other hand, our framework can be trained without ROI annotations, while still being able to localise the most informative parts of the images. Moreover, the creation of saliency maps is an integral part of our pipeline. By explaining the inner workings of a neural network, saliency maps can increase trust in the model's predictions and support the decisions of clinicians.

## 7 REPRODUCIBILITY STATEMENT

Code to reproduce the experiments will be shared with the reviewers and area chairs during the discussion phase. Training details, such as hyperparameters chosen and data splits are included in Appendix C, together with information related to the computing resources. The architectures of the models employed are presented in Appendix B and additional details about the experimental setup are provided in Section 4.2. Information about the datasets used in the experiments can be found in Appendix A.

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

## A  DATA

**IST-3** or the Third International Stroke Trial is a randomised-controlled trial that collected brain imaging (predominantly CT scans) from 3035 patients with stroke symptoms at two time points, immediately after hospital presentation and 24-48 hours later. Among other things, radiologists registered the presence or absence of early ischemic signs. For positive scans, they also coded the lesion location. In our experiments, we only employed the labels for the following classes: no lesion, lesion in the left side, lesion in the right side, lesion in both sides of the brain. $46.31\%$ of the scans we considered are negative and the remaining are positive. In particular, $28.80\%$ have left lesion, $24.03\%$ right lesion and $0.86\%$ lesion in both sides of the brain. The information related to the more specific location of the lesion was only employed to test the score of the saliency maps presented in Section 3.1 and never used at training time. Further information about the trial protocol, data collection and the data use agreement can be found at the following url: IST-3 information.

**MosMed** contains anonymised lung CT scans showing signs of viral pneumonia or without such findings, collected from 1110 patients. In particular, $40.4\%$ of the images we conisdered are positive and $59.6\%$ are negative. In a small subset of the scans, experts from the Research and Practical Clinical Center for Diagnostics and Telemedicine Technologies of the Moscow Health Care Department have annotated the regions of interest with a binary mask. However, in our experiments we didn't employ these masks. Further information about the dataset can be found in Morozov et al. (2020).

## B  ARCHITECTURES

The MTL model classifies whether a brain scan has a lesion (is positive) or not. If the scan is positive, it also predicts the side of the lesion (left, right or both). In order to do so, a MTL CNN with 7 convolutional layers and two classification heads is employed. In the first stage, the CNN considers only half scans (left or right) and processes one slice of each scan at a time. Then, the extracted

features from each side are concatenated and averaged across the slices of each scan, before reaching the two classification heads. The classification accuracy is computed considering whether the final classification output identifies the correct class out of the four possible or not. In the ResNet-50 architecture used for the classification of lung CT scans, we still process one slice at a time and average the slices before the classification layer. In particular, we performed a binary classification task between scans with with moderate to severe COVID-19 related findings (CT-2, CT-3, CT-4) and scans without such findings (CT-0). The autoencoder used to reconstruct images has 3 ResNet convolutional blocks both in the encoder and in the decoder parts, with $3 \times 3$ filters and no bottleneck.

## C  TRAINING DETAILS

The baseline models were trained for 200 epochs and then employed, together with an autoencoder trained to reconstruct the images, to obtain the saliency maps that are needed for our framework. Our framework and the competing methods were fine-tuned for 100 epochs, starting from the weights of the baseline models. The training procedure of ACAT is summarised in Algorithm 1.

---

**Algorithm 1:** ACAT training

**Data:** $\mathcal{D} = (x^i; i = 1, 2, \ldots, N_D)$
1 Train baseline classification network $f$ and autoencoder $D(E)$ on $\mathcal{D}$
2 Given $E(x^j) = z^j$, minimise: $g(z) = L(d(z), t) + \alpha||z - E(x^j)||_{L_1}$
3 Decode the obtained latent vector to compute the counterfactual $D(z')$
4 Obtain saliency maps $S^j$ from positive and negative counterfactuals
5 Train ACAT on $\mathcal{D}$ using $x^j$ and $S^j$ as input

---

In the case of IST-3 data, we uniformly sampled 11 slices from each scan and resized each slice to $400 \times 500$, while for MosMed data we sampled 11 slices per scan and then resized each slice to $128 \times 128$. All the networks were trained using 8 NVIDIA GeForce RTX 2080 GPUs. For each model, we performed three runs with different dataset splits, in order to report average accuracy and standard error.

## D  COMPETING METHODS FOR SALIENCY-AIDED CLASSIFICATION

In the saliency-modulated image classification (SMIC) (Flores et al., 2019), the branch that is used to pre-process the saliency maps has two convolutional layers. For the other implementation details, we follow Flores et al. (2019). For SalClassNet (Murabito et al., 2018), we tried to follow the original implementation by using the saliency maps generated with our approach as targets for the saliency branch, since we don't have the ground-truth saliency maps available, but this led to poor results. For this reason, rather than generating the saliency maps with the saliency branch, we compute them with our approach. Then, as in Murabito et al. (2018) we concatenate them with the input images along the channel dimension. For the hallucination of saliency maps (HSM) approach, following Figueroa-Flores et al. (2020), the saliency detector has four convolutional layers. In SpAtt we consider a network with only one branch and compute the soft spatial attention masks directly from the image features, at the same stage of the network where saliency attention masks are computed in our framework. SeAtt employes self-attention modules from Zhang et al. (2019), which are placed after the third and fifth convolutional layer in the MTL architecture and after the third and fourth convolutional block in the ResNet-50.

## E  FAILURE MODES OF COMPETING METHODS FOR THE GENERATION OF COUNTERFACTUALS

We observed that in several cases, when generating counterfactual examples, the latent shift method is not able to achieve low values for the probability of the class of interest $p$. We show here two examples of positive brain scans, for which we attempt to generate counterfactuals with low probability of lesion according to the classifer $f$, starting from a probability close to 1 . We apply one-step gradient

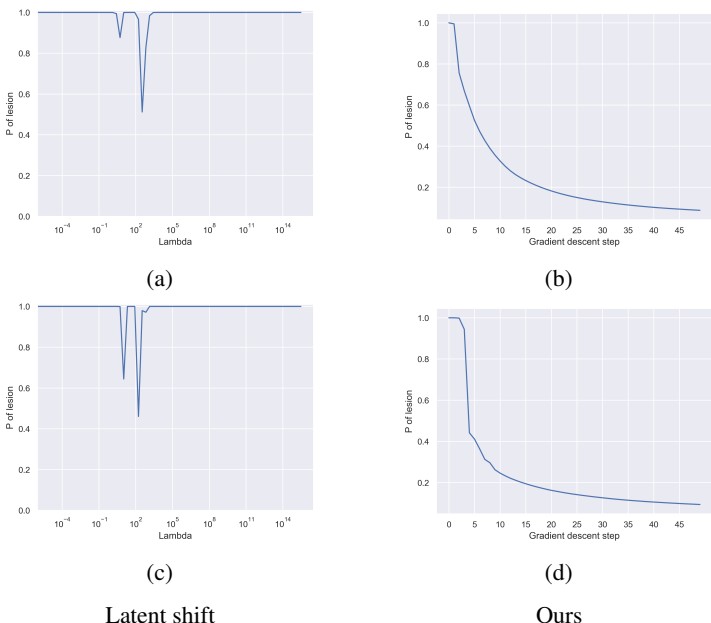

(a)                                     (b)

(c)                                     (d)

Latent shift                            Ours

Figure 5: Probability of lesion obtained with one step-gradient updates in the latent space (Cohen et al., 2021) for different values of the step size $\lambda$ for two samples ((a) and (c)) and with gradient descent minimising Eq. (2) ((b) and (d))

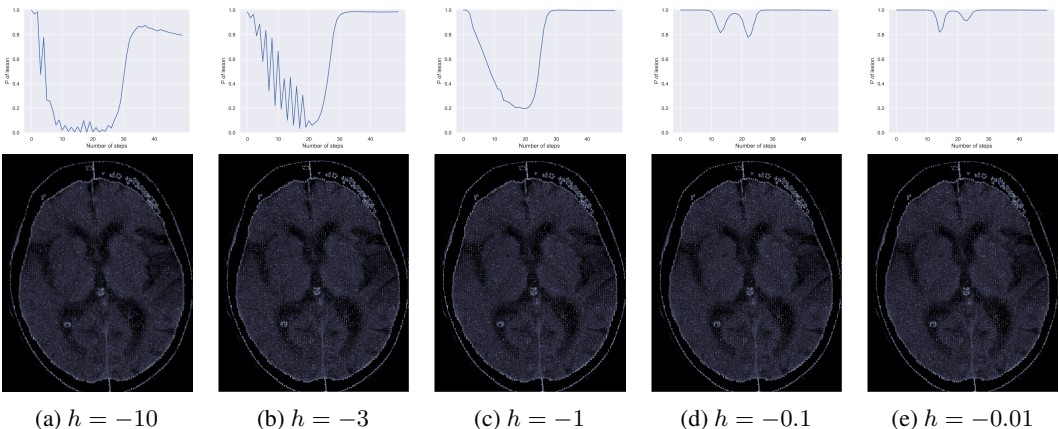

(a) $h = -10$          (b) $h = -3$          (c) $h = -1$          (d) $h = -0.1$          (e) $h = -0.01$

Figure 6: In the top panel are shown the probability of lesion obtained with progressive gradient updates in the latent space, with the step size value fixed to -10 (a), -3 (b), -1 (c), -0.1 (d), -0.01 (e) and no bound on the latent move. In the bottom panel are displayed the counterfactual examples obtained at the gradient step where $p$ is minimal

updates as in Cohen et al. (2021), starting with the step size value $\lambda = 1e - 5$ and multiplying $\lambda$ by two at each successive attempt. In Figure 6(a) and (c), we show the probability of lesion as a function of $\lambda$ for these two samples. We can observe that the minimum value obtained for $p$ is $0.51$ for the first sample and $0.46$ for the second one. On the other hand, by following our approach and minimising Eq. (2) by gradient descent, with target class 'no lesion', $p$ reaches a value lower than $0.2$ with 20 gradient updates in both cases and then converges to 0 (Figure 5(b) and (d)). In these runs we employed a step size of 1. However, different step sizes yield similar results for the probability functions.

For the first sample, we also test a method where we perform small progressive updates of size $h$ in latent space, but without a bound on the distance between original and counterfactual images. $P$

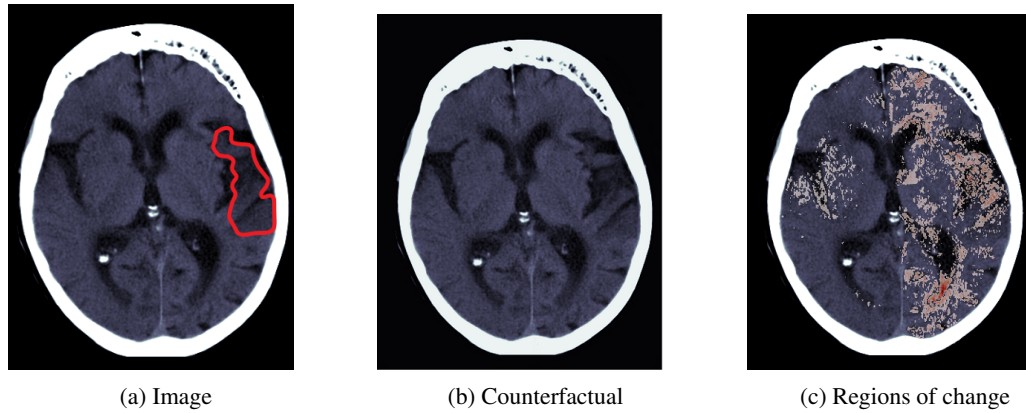

| (a) Image | (b) Counterfactual | (c) Regions of change |

Figure 7: Counterfactual example with $p = 0.08$ generated with our approach (b) and regions of change (c), with respect to the original image (a), highlighted with a red color map. The regions of change have a good overlap with the area of the lesion indicated with the red arrow in (a).

of the resulting images is shown in Figure 6 for values of $h$ in $\{-10, -3, -1, -0.1, -0.01\}$. With $h = -10$, $h = -3$ and partially with $h = -1$, we are able to reach low values of $p$, but the probability function has an unstable behaviour and later starts increasing, rather then converging to 0. With the other values of $h$, we are never able to achieve low values of $p$. The graphs are shown in the top panel of Figure 6. The counterfactual images obtained at the gradient update steps where $p$ is minimal in these optimisation runs, are showed in the bottom panel of the same Figure. In all cases, the images largely differ from the original brain scan, displayed in Figure 7(a) and are not semantically meaningful. On the other hand, with our approach we are able to obtain a credible counterfactual, displayed in Figure 7(b), together with its regions of change with respect to the original image 7(c). We can observe that the regions of change largely overlap with the area of the lesion highlighted with the red arrow in Figure 7(a), suggesting that the counterfactuals generated with our approach are semantically meaningful.

## F    FURTHER EVALUATION OF SALIENCY MAPS

In Section 4.3 we observed how the saliency maps generated with Grad-CAM obtain a poor score. We test if more recent improvements of the method can have a significant impact on the score obtained. In particular, we considered Grad-CAM++ (Chattopadhay et al., 2018) and Score-CAM (Wang et al., 2020). The former, in order to provide a measure of importance of each pixel in a fetaure map for the classification decision, introduces pixel-wise weighting of the gradients of the output with respect to a particular spatial position in the final convolutional layer. On the other hand, the latter removes the dependence on gradients by obtaining the weights of each activation map through a forward passing score for the target class. We observed that Grad-CAM++ very marginally improves the performance of Grad-CAM (from $11.67\%$ (1.28) to $11.78\%$ (0.46)), while Score-CAM obtains the worst score with $9.90\%$ (0.78). Finally, we also tested the Integrated Gradient method (Sundararajan et al., 2017), in which the gradients are integrated between the input image and a baseline image, achieving a score of $37.52\%(4.11)$. These methods obtain scores that are considerably lower than the ones of adversarial approaches.

## G    VISUALISATION OF COUNTERFACTUAL EXAMPLES

In Figure 8, we display the counterfactual examples of the images displayed in Fig. 3, obtained with our approach and the latent shift method. Saliency maps of the change are displayed in Figure 3.

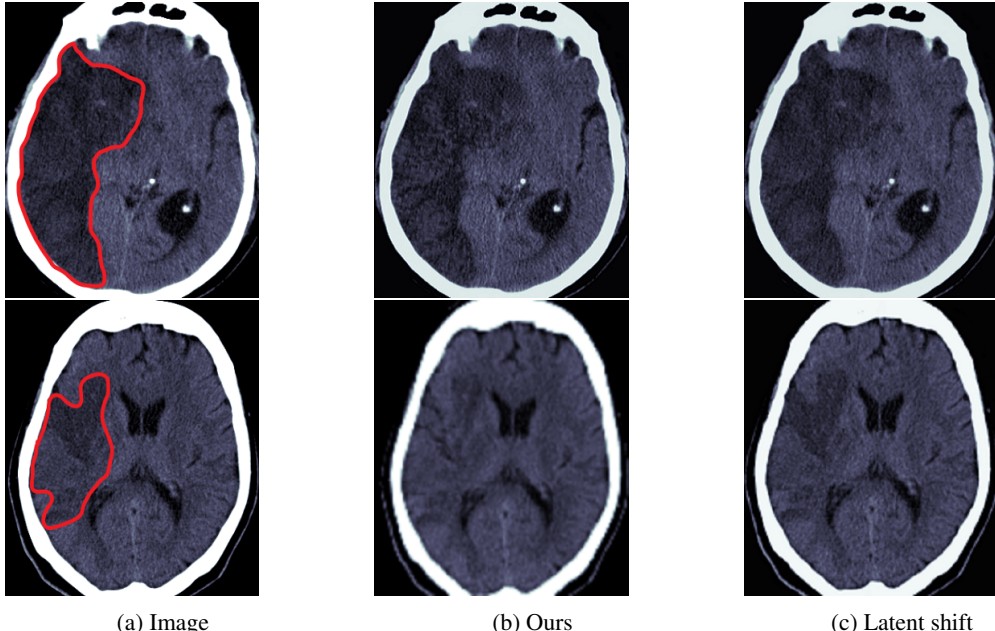

| (a) Image | (b) Ours | (c) Latent shift |

Figure 8: (a) Ischaemic stroke lesion appears darker than normal brain. Counterfactual examples for the negative class obtained with our approach (b) and the latent shift method (c)

## H    IoU and Dice score of saliency maps

We compared the proposed method against competing saliency generation approaches, including the latent shift method and progressive gradient descent updates but with no reconstruction loss or limitation of the move in the latent space (NoRec). In particular, we considered 50 test samples in the MosMed dataset for which annotation masks are available and evaluated the IoU score (Jaccard Index) and the Dice coefficient (F1 score). Following Cohen et al. (2021) and Viviano et al. (2019), we binarized the saliency maps by setting the pixels in the top p percentile to 1, where p is chosen dynamically depending on the number of pixels in the ground truth it is being compared to. The results are shown in Table 3. Out of the methods considered, our approach achieves both the best IoU and Dice coefficient (0.5203 and 0.5372 respectively). NoRec slightly improves the scores obtained with the latent shift method.

Table 3: Dice coefficient and IoU score computed on 50 test scans on MosMed to compare different saliency generation approaches. Our approach achieved the best score in both evaluation metrics

|  | IoU | Dice |
|---|---|---|
| Gradient | 0.5022 (0.0005) | 0.5071 (0.0009) |
| Grad-CAM | 0.4998 (0.0003) | 0.5024 (0.0006) |
| Latent shift | 0.5116 (0.0005) | 0.5241 (0.001) |
| NoRec | 0.5138 (0.0022) | 0.5260 (0.0008) |
| Ours | **0.5203** (0.001) | **0.5372** (0.0012) |

## I    Ablation studies on ACAT architecture

We have performed further ablation studies to test our architecture. In the proposed approach, attention masks are obtained from the saliency branch at three different stages of the network (early, middle and late) and finally an attention fusion layer weighs the three masks and is applied before the classification layers. Therefore, we progressively removed the fusion layer, the late attention mask and the middle attention mask to test the contribution of each component. While the classification accuracy of the full ACAT architecture on MosMed was $70.84\%(1.53)$, by removing the attention

fusion layer it decreased to $69.79\%(2.78)$. Moreover, by also removing the late attention layer it further decreased to $68.75\%(1.48)$, reaching $68.23\%(0.85)$ when the middle attention layer was eliminated as well.

