# OpenReview forum: "ACAT: Adversarial Counterfactual Attention for Classification and Detection in Medical Imaging"
_ICLR.cc/2023/Conference — Submitted to ICLR 2023_

### Official Review · Reviewer_3Yeb · 2022-10-24

**Confidence:** 5
**Clarity, Quality, Novelty And Reproducibility:** the originality is there, but novelty…
**Correctness:** 3
**Technical Novelty And Significance:** 2
**Empirical Novelty And Significance:** 2
**Recommendation:** 3

**Strength And Weaknesses:**

Strengths:
--The idea using counterfactual images for saliency map generation is interesting.

--The improvement for medical imaging taks is significant.

Weaknesses:

--The novelty is simple and limited.

--More experiments are needed, such as existing counterfactual generation.

**Summary Of The Paper:**

This paper proposes an attention generation method for ROI detection by adversarial counterfactual without attention label. The attention map can be used to highlight useful information for disease classification and detection. The experiments show its improvements on different medical imaging tasks.

**Summary Of The Review:**

the proposed method is interesting, but the novelty is limited

---

> ### Author Response · Authors · 2022-11-18
> **Response to reviewer 3Yeb**
>
> Unfortunately, the review is extremely short and vague. We find it difficult to provide a detailed rebuttal. The main criticisms are formulated through general statements that we cannot relate to specific parts of the paper, like  “some claims have minor issues”, “a few statements”.
>
> We also believe that if the improvements obtained on the medical imaging tasks are significant, our work could be interesting for the community. We are available to respond to further, more specific concerns.
>
> Code has now been shared with reviewers.

---

### Official Review · Reviewer_svqR · 2022-10-24

**Confidence:** 4
**Correctness:** 3
**Technical Novelty And Significance:** 2
**Empirical Novelty And Significance:** 3
**Recommendation:** 5

**Clarity, Quality, Novelty And Reproducibility:**

The counterfactual saliency map technique is clearly motivated and described, as are the datasets and experiments. The main technical novelties appear the progressive optimization of the latent shift method, and the integration of saliency map attention at multiple scales.


**Strength And Weaknesses:**

Strengths:

1. Presents a method for improving classification performance by automatically integrating saliency map information.

2. Compares against several existing saliency map fusion methods.

3. Saliency maps of the various methods visualized.

Possible Weaknesses/Considerations:

The paper appears to contain two main contributions: regularized latent shift to produce counterfactual images, and thus saliency maps, and the use of these saliency maps to guide classification via attention. While some ablation experiments are attempted with other saliency map methods (e.g. Grad-CAM), these experiments might not be fully complete.

4. Segmentation performance with ground truth annotations is reported for the regularized latent shift method against other saliency map methods, in Section 4.3 and Appendix H. To begin with, the attribution map evaluation in the main text appears to consider only “the pixel with the greatest value in each CT scan”, with the annotated region. It is not clear if the consideration of just a single pixel (albeit of greatest value) in an entire CT scan to evaluate the performance of a saliency map, is particularly meaningful.

5. IOU and Dice score performances are then reported in Appendix H, but details such as how each saliency map is thresholded do not appear to be described. Moreover, the set of saliency map methods tried in Appendix H appear not entirely the same as in Section 4.3 (NoRec is missing).

6. The set of saliency map methods tested does not appear very comprehensive; more recent methods such as Integrated Gradients do not appear included.

7. The contribution of different saliency map methods to classification performance (as reported in Tables 1 & 2) does not appear to be evaluated directly, as Table 1 only reports performance against other frameworks.

8. For classification performance, only accuracy is reported. It would be highly recommended for common metrics such as sensitivity, specificity and AUROC to be included, to better reflect the performance of the various classifiers over the full range of possible thresholds.

9. The contribution of multiple attention masks at different scales (Section 3.2.1), instead of at a single scale, does not appear directly evaluated.



**Summary Of The Paper:**

This paper proposes Adversarial Counterfactual Attention (ACAT), which first generates regions of interest (ROI) in (medical) images with saliency (heat)map techniques, and then uses these saliency map ROIs to generate soft spatial attention masks at various scales, that are integrated into a deep learning classifier. The counterfactual images are generated using small progressive shifts in the latent space, to encourage a smooth transition from the original image to the counterfactual. ACAT is reported to increase the baseline classification accuracy of lesions and COVID-19 related findings, from CT scans.


**Summary Of The Review:**

This paper proposes an Adversarial Counterfactual Attention (ACAT) method that uses counterfactual saliency maps to guide classifiers through attention. Some uncertainty remains as to whether the proposed counterfactual saliency map method is optimal for the classification task, and as to the evaluation of the saliency maps on annotations.

---

> ### Author Response · Authors · 2022-11-18
> **Response to reviewer svqR**
>
> We thank reviewer svqR for the suggestions. In order to evaluate the saliency maps on IST-3 data (section 4.3), we only considered the pixel with the highest change because for this dataset we don’t have the ground truth annotations of the lesions, but we only know the region where they are located. Moreover, the highest intensity pixel of the saliency map is the pixel with the biggest variation between the original image and the counterfactual example. Therefore, we would expect it to be located in the area of the lesion. IoU and Dice scores could only be computed on MosMed data, where we have annotation masks for 50 samples.
>
> In the first analysis, we didn’t consider NoRec, because we don’t expect a regularisation term minimising the change from the original image to impact the single highest intensity pixel. On the other hand,  this regularisation term can have effects on the overall saliency map and therefore it was included in the IoU and Dice score analysis. As mentioned in Appendix H, we binarized the saliency maps by setting the pixels in the top p percentile to 1, where p is chosen dynamically depending on the number of pixels in the ground truth it is being compared to.
>
> We thank the reviewer for the suggestion to also test saliency maps obtained with the integrated gradient method. We followed the implementation of the original paper by  Sundararajan et al. [1] and evaluated the saliency maps with the score presented in Section 4.3. The score obtained with this approach on IST-3 is 37.52% (4.11) and therefore worse than the ones obtained with the gradient method, the latent shift and our approach.
>
> The contribution of different saliency maps method on classification performance, given a certain fixed architecture (ACAT), is studied in Section 4.5, where we observe that the saliency generation approaches obtain the same ranking that was achieved with the direct scoring of saliency maps performed in Section 4.3.
>
> As mentioned in the previous response, we also tested the impact of progressively removing the fusion layer, the late attention mask and the middle attention mask to test the contribution of each component.
> While the classification accuracy of the full ACAT architecture on MosMed was 70.84\% (1.53), by removing the attention fusion layer it decreased to 69.79\% (2.78). Moreover, by also removing the late attention layer it further decreased to 68.75\% (1.48), reaching 68.23\% (0.85) when the middle attention layer was eliminated as well.
>
> References:
>
> [1] Sundararajan, Mukund, Ankur Taly, and Qiqi Yan. "Axiomatic attribution for deep networks." International conference on machine learning. PMLR, 2017.

---

### Official Review · Reviewer_McTW · 2022-10-28

**Confidence:** 4
**Correctness:** 3
**Technical Novelty And Significance:** 2
**Empirical Novelty And Significance:** 2
**Recommendation:** 5

**Clarity, Quality, Novelty And Reproducibility:**

### Clarity:
The article is globally clear (minus a summary of the method, cf weaknesses) and the related work is good.

### Novelty:
I fell that the novelty is a bit missing

### Reproducibility:
The method seems easy to implement and the authors indicated that code would be available.

**Strength And Weaknesses:**

### Strengths:
- the method is simple (no iteration on the adversarial parts)
- the shown results are good
- good related works

### Weaknesses:
- I think that the main weakness of the method is its novelty: in Flores et al., saliency maps where already generated using an adversarial approach with an auto-encoder and the contributions on this part seems to be the use of several smaller gradient steps instead of one ¨large" step.
- Similarly, the network architecture is similar to ones previously used to incorporate saliency maps but with a multi-scale approach. Once again, the results are better with this approach but when a proposed model is a small improvement over a previous one, I feel that more experiments are required to add value to the article: for example, is it really using a multi-scale model that helps or the fact that the smaller scale can detect small RoI? (as suggested by the drop of performance for large RoI).
- the dropout experiment if I understand it correctly is not fully convincing. If the dropout is not spatially constrained, information from all parts of the image still go through even with p=0.5. Maybe a layer with a random cutout (or RandomErasing in torch) might be more similar to a RoI approach.
- Maybe adding a small ¨Algorithm box" with the inputs and outputs of both steps would be good (First step: pretrained model and auto encoder -> RoI detector, 2nd step: baseline + RoI detector -> final classifier)


**Summary Of The Paper:**

The paper proposes a method based on saliency maps to guide a classifier in tasks where only part of the image is relevant. The saliency maps are generated using an adversarial gradient descent on a fixed classifier and autoencoder. They are then combined with the classifier at several scales to finetune it. The method is validated on 2 datasets.

**Summary Of The Review:**

My initial recommendation tend towards rejecting this work: while the results are good, I feel that the novelty of the proposed method is not sufficient and that more ablations studies to fully understand the model are necessary to accept it.

---

> ### Author Response · Authors · 2022-11-18
> **Response to reviewer McTW**
>
> We thank reviewer McTW for the useful comments.
>
> We have performed further ablation studies to test our architecture on MosMed data. In the proposed approach, attention masks are obtained from the saliency branch at three different stages of the network (early, middle and late) and finally an attention fusion layer weighs the three masks and is applied before the classification layers. Therefore, we progressively removed the fusion layer, the late attention mask and the middle attention mask to test the contribution of each component. While the classification accuracy of the full ACAT architecture on MosMed was 70.84\% (1.53), by removing the attention fusion layer it decreased to 69.79\% (2.78). Moreover, by also removing the late attention layer it further decreased to 68.75\% (1.48), reaching 68.23\% (0.85) when the middle attention layer was eliminated as well.
>
> We thank the reviewer for the suggestion to test random cutout. In particular, we applied it to the baseline network on the image features at the same layers where the attention masks are applied in ACAT. The results obtained were worse than the baseline performance, obtaining 66.51% and 61.48% accuracy on IST-3 for probability of cutout respectively 0.2 and 0.6 and 63.02% and 62.50% accuracy on MosMed for the same values of p.
>
> We  also added an algorithm box summarising the training process of ACAT in Appendix C.

---

### Decision · Program_Chairs · 2023-01-20

**Decision:**

Reject

**Justification For Why Not Higher Score:**

The method is mainly empirical, and has similarity to other published methods that use saliency in image analysis.  "Small" improvements on specific medical imaging tasks without substantial theoretical grounding or higher methodological novelty are more appropriate for publication in a medical venue.

**Justification For Why Not Lower Score:**

N/A

**Metareview: Summary, Strengths And Weaknesses:**

The submission proposes a method for using saliency maps to guide a classifier in tasks where only part of an image is relevant.  The method has similarities to Flores et al., cited in the paper, but with some engineering innovations that lead to somewhat better performance than competing methods in the same family, specifically the use of the regularized latent shift method vs. other saliency map methods.  Multiple reviewers raised concerns about novelty and had concerns about missing ablation studies.  The latter points about ablations were reasonably addressed in the rebuttal, improving the manuscript.  On the balance, though, the submission has two main issues that bring into question its appropriateness for ICLR in its current form: 1) the novelty is indeed somewhat limited, certainly as a take-home message for a wider ICLR audience, and 2) the results are primarily empirical with little theoretical grounding or guarantees.  It is true that the empirical results seem to be better on specific medical imaging tasks, albeit by a couple percent.  That is a statistically significant improvement in most cases, but the submission may be better targeted towards a venue such as MICCAI than ICLR, given that the main outcome is better performance on medical image analysis tasks.